# Evaluating the Capability of Sentinel-1 Data in the Classification of Canola and Wheat at Different Growth Stages and in Different Years

Lingli Zhao [1], Shuang Wang [1], Yubin Xu [2], Weidong Sun [1,*], Lei Shi [3], Jie Yang [3] and Jadunandan Dash [4]

[1] School of Remote Sensing and Information Engineering, Wuhan University, Wuhan 430079, China; zhaolingli@whu.edu.cn (L.Z.)

[2] China Academy of Civil Aviation Science and Technology, Beijing 100028, China

[3] State Key Laboratory of Information Engineering in Surveying, Mapping and Remote Sensing, Wuhan University, Wuhan 430079, China

[4] School of Geography and Environmental Science, University of Southampton, Southampton SO17 1BJ, UK

* Correspondence: widensun2012@whu.edu.cn

**Abstract:** Canola and wheat are the main oilseed crop and grain crop, respectively, and they often have similar phenological stages. The understanding of the interactions between microwave signals with wheat and canola in different stages is important for their monitoring using synthetic aperture radar (SAR) imagery. This paper investigates the characteristics of canola and wheat through the use of backscattering profiles from multi-year Sentinel-1 images. Large fluctuations are observed for the temporal backscattering profiles of canola and wheat in different growth statuses induced by agrometeorological conditions in different years. The capability and stability of Sentinel-1 for wheat and canola mapping is further investigated using single- and multi-temporal SAR images hosted in Google Earth Engine (GEE) using the random forest classifier. Although different agrometeorological conditions and field managements make the temporal profiles of backscattering variations, the large difference in canopy structure allows SAR images to make the separability of canola and wheat stable on Sentinel-1 images in different phenology stages. The classification accuracies and the feature importance scores from multi-temporal classification in different years show that the backscattering features obtained at flowering to maturity stages make more contributions to the good-quality mapping of canola and wheat than those at other stages. The F1 scores of canola and wheat achieve 0.95 during the canola flowering and podding period, and the minimum F1 scores of 0.85 were also obtained at other stages. These findings show that SAR images have great potential in the good-quality mapping of canola and wheat in a wide phenology window.

**Keywords:** synthetic aperture radar; wheat; canola; Sentinel-1; Google Earth Engine

## 1. Introduction

The timely monitoring of canola and wheat crops, which are the world's main oilseed and grain crop, respectively [1,2], is critical to the balance of food supplies and food policy making [3]. Canola and wheat are often planted at the same time and have similar growth cycles in many areas. Their mapping on a large scale is the prerequisite for area extraction, yield prediction and agricultural disaster prevention. Remote sensing techniques are of great value to the effective management and monitoring of the environmental resources of crops by providing frequent observations on a large scale [4–9]. Optimal observation time has been studied for the discrimination of canola and wheat using different vegetation indices derived from multispectral images [10]. It has been found that the spectral signature exhibited during the flowering of canola can be used to effectively separate canola from other crops [11–13]. Ashourloo et al. [11] developed an efficient index to separate canola from wheat, based on their reflectance difference in near-infrared, green and red bands

at the flowering stage. The normalized difference yellow index was also developed to discriminate canola from wheat-like crops at the flowering stage [12,13]. It is important to obtain multi-spectral images during the flowering period of canola for its mapping. However, the time window of optical images is narrow and has no guarantee due to the effect of cloudy and rainy days.

Synthetic aperture radar (SAR) provides the unique opportunity of regular observations for crop management, due to its capability of evading the effects of rain and clouds [14–17]. In particular, changes in the geometric structures and dielectric properties of crop canopies at different growth stages induce changes in microwave backscattering [18–20]. The mapping capability of some crops has been illustrated using different classification frameworks such as machine-leaning- [21,22] or polarimetric-decomposition-based classifiers [23–25]. These results show that backscattering coefficients, polarimetric information and image acquiring time are all critical to achieve good classification results.

The backscattering of crops is greatly affected by canopy, especially the shape and density of leaves and the structure of stems. The capability of SAR to distinguish broad-leaved and narrow-leaved crops has been proven by researchers [14,26,27]. Arias et al. [14] identified crops such as barley and corn with different canopy structures using time series Sentinel-1 data. Wiseman et al. [26] found that there was a strong correlation between polarimetric responses and dry biomass for canola and corn with broad leaves, but a weak correlation for wheat with narrow leaves. Gella et al. [27] improved crop classification accuracy by merging grain crops with similar leaf geometry by using multi-temporal SAR images. Wheat and canola are two typical crops with different leaf shapes and canopy structures, and some features on radar images have also been investigated for them in specific phenological stages. High scattering randomness is observed for canola over the pod development and ripening stages when pods in canopy increase the multiple scattering [24,28]. Moreover, differential attenuation was observed over wheat, whose vertical structure is significant at the jointing stage [29,30]. Some endeavors have been made to find stable temporal profiles for canola and wheat and to analyze the sensitivity of radar responses to their growth dynamics [29–31]. Bhogapurapu et al. [31] observed that there are similar morphological changes during the growth of canola and wheat using Sentinel-1 data. Mandal et al. [32] derived a dual-polarization radar vegetation index called DpRVI that has a good positive correlation with the leaf area index, vegetation water content and dry biomass of canola. Schlund and Erasmi [33] detected the shooting and harvesting stages of wheat using the smoothed cross-polarization ratio of Sentinel-1 temporal images to reduce the effect of underlying surface.

Although attempts have been made to characterize and map canola and wheat using single- and multi-temporal SAR images, few studies have explored the classification capability of SAR images for wheat and canola at different phenological stages and the stability of SAR image classification under different agrometeorological environments. There are still considerable variations in the SAR observations, due to the effect of management activities and environmental stresses in different years. For example, crops have different growth statuses when rainfall, temperature or sowing date change in different years. In this paper, time series Sentinel-1 images were acquired from 2016 to 2019 over a canola and wheat production area in China to (1) characterize the temporal backscattering profiles of canola and wheat and highlight their main evolution trends and fluctuation, and (2) further explore the capability of Sentinel-1 for the mapping of canola and wheat in different phenological stages and in different years.

## 2. Study Area and Dataset

### 2.1. Study Area

The study area is located in Erguna, Inner Mongolia, China, as shown in Figure 1, and this region is the main spring-wheat- and rapeseed-producing area in China. The main crops in the area are canola and wheat, and the cropland parcels of the farmland range from 0.33 km$^2$ to 5.0 km$^2$ in area. Although the cultivation methods are all machinery-based for

the farmland, the management activity takes a long time due to the low population. For example, the sowing period of crops may last for half of one month, and the growth cycle of wheat and canola is similar in the study area. The rough growth schedule of canola and wheat is shown in Figure 2: they are both sown from April and harvested in late August. There is crop rotation in the study area in different years. The crop type, sowing date and ploughing date of the fields in different years were provided by the farmland managers.

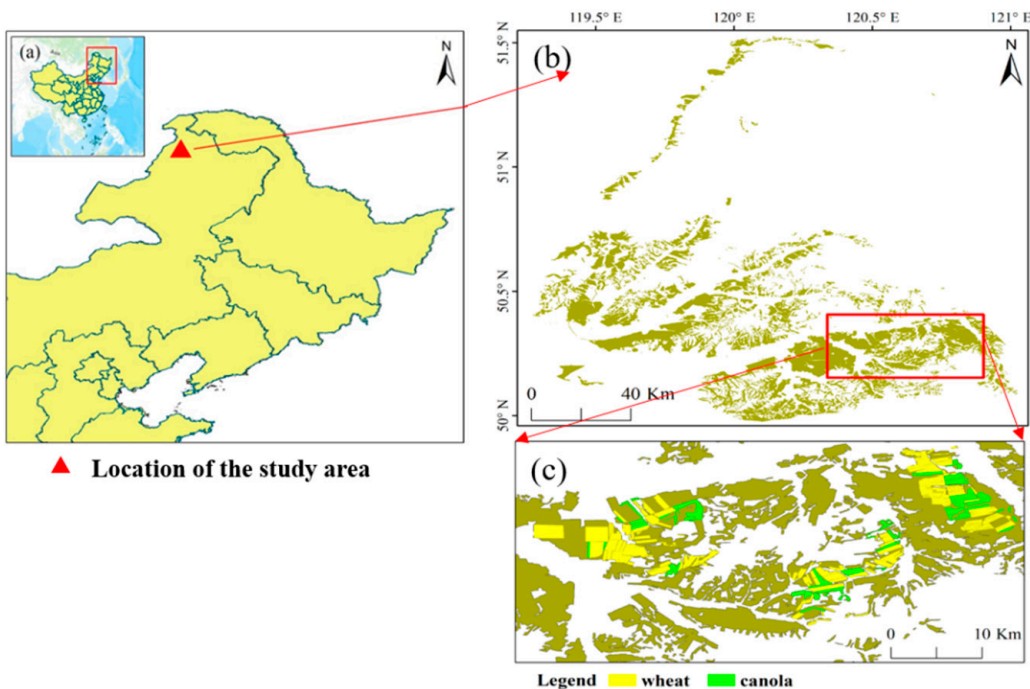

**Figure 1.** (**a**): Location of the study area, (**b**) farmland of study area derived from land cover data, (**c**) ground truth of Erguna in 2019, provided by farmland managers.

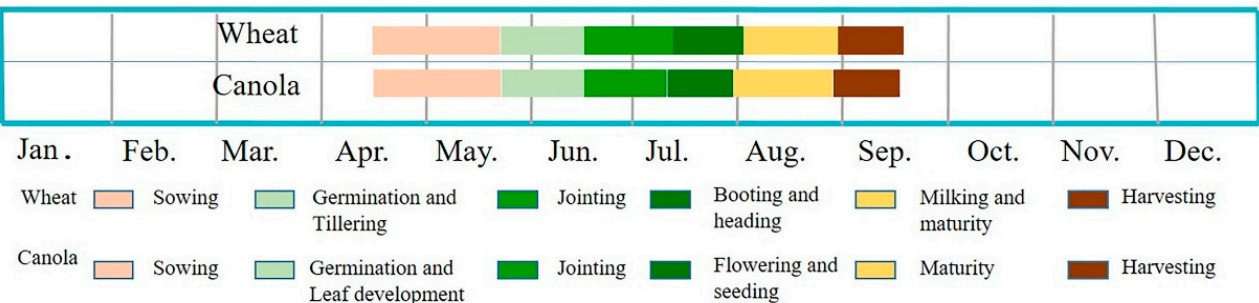

**Figure 2.** Rough growth calendar of canola and wheat in Erguna.

Most of the fields are planted with rainfed crops, and there is no manual irrigation during the growth cycle. The daily accumulative temperature and rainfall that were collected from the China Meteorological Data Service Center (http://data.cma.cn/, accessed on 1 June 2022) are shown in Figure 3. It shows that there were large changes in the agrometeorological conditions in different years, which resulted in large fluctuations in yields of canola and wheat from 2016 to 2019, as shown in Figure 4. There were favorable weather conditions for a bumper harvest in 2018, whereas in 2017, a severe drought event occurred in the growth stage of canola and wheat, resulting in yield reduction. Different meteorological conditions coupled with crop rotation lead to variations in soil properties and crop distribution in Erguna.

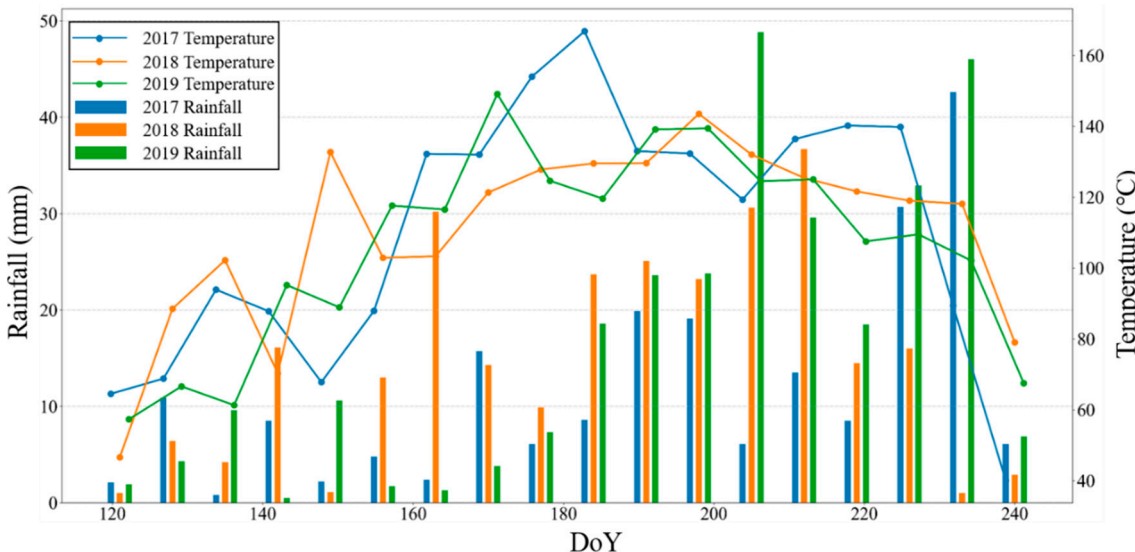

**Figure 3.** Cumulative rainfall and temperature by week in Erguna between 2017 and 2019, from May to August.

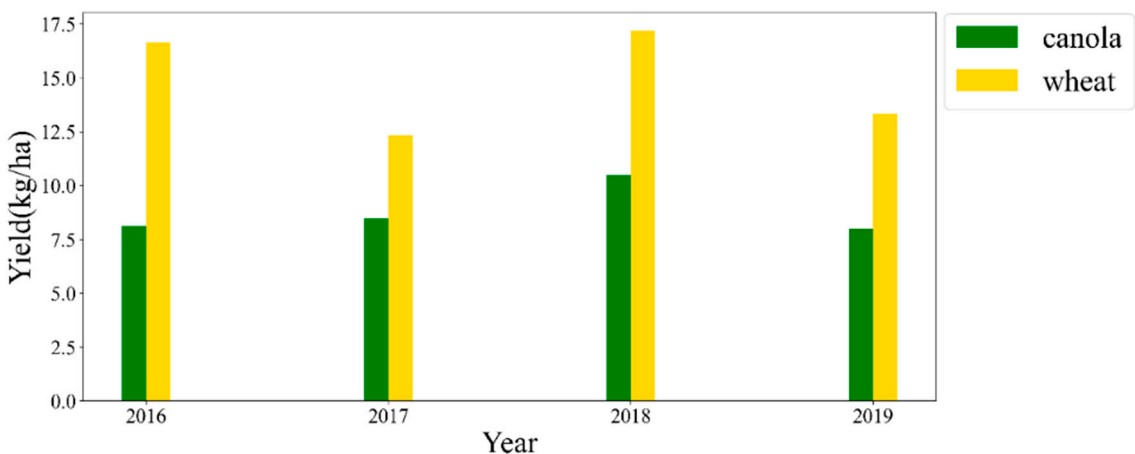

**Figure 4.** Yields of canola and wheat in Erguna from 2016 to 2019.

*2.2. SAR Data*

The Sentinel-1 Ground Range Detected (GRD) products acquired over the growth period of the years from 2016 to 2019 were used. The revisit time of Sentinel-1 is 12 days, and the acquired images are shown in Table 1. The central frequency of Sentinel-1 is 5.405 GHz (C band), and the dataset was obtained in interferometric wave mode (IW) with a ground resolution of 10 m [34]. The time series Sentinel-1 images were acquired and processed in the cloud-based Google Earth Engine (GEE, https://earthengine.google.com/, accessed on 1 July 2022) platform to improve processing efficiency. This Sentinel-1 collection hosted in GEE was preprocessed using the following steps: updating orbit metadata, thermal noise removal, radiometric calibration and terrain correction. The Refined Lee filter [35] was then applied to all scenes with a window size of $7 \times 7$ to suppress the speckle noise. Then the filtered images were converted to decibels via log scaling ($10 \times \log10$ (image)). One thing to note is that the Sentinel-1 image of 18 June 2016 was missing, and the backscattering coefficients were obtained by linear interpolation of the adjacent images for temporal profile analysis.

**Table 1.** Parameters of the used Sentinel-1 images.

| Year | Mode | Polarization | Date of Acquisition | Orbit | Incidence Angel (°) |
|------|------|--------------|---------------------|-------|---------------------|
| 2016 | IW | VV + VH | 19 Apr., 1 May, 13 May, 25 May, 6 Jun., 30 Jun., 12 Jul., 24 Jul., 5 Aug., 17 Aug., 29 Aug., 10 Sep. | Descending | 36–44 |
| 2017 | IW | VV + VH | 2 May, 14 May, 26 May, 7 Jun., 19 Jun., 1 Jul., 13 Jul., 25 Jul., 6 Aug., 18 Aug., 30 Aug., 11 Sep. | Descending | 36–44 |
| 2018 | IW | VV + VH | 27 Apr., 9 May, 21 May, 2 Jun., 14 Jun., 26 Jun., 8 Jul., 20 Jul., 1 Aug., 13 Aug., 25 Aug., 6 Sep. | Descending | 36–44 |
| 2019 | IW | VV + VH | 22 Apr., 4 May, 16 May, 28 May, 9 Jun., 21 Jun., 3 Jul., 15 Jul., 27 Jul., 8 Aug., 20 Aug., 1 Sep., 13 Sep. | Descending | 36–44 |

### 2.3. Auxiliary Data and Ground Reference Data

The land cover map of GlobeLand30 [36] was used to extract the farmland of Erguna to mask out other landcover types; the spatial resolution of this dataset was 30 m and the overall accuracy of the dataset was 85.72%. In order to further suppress SAR speckle noise and to reduce the heterogeneity within field, an object-oriented classification method was applied in the experiment. Considering that the boundary of the field is clear on the optical images, the Sentinel-2 Top of Atmosphere (TOA) data of the study area were used to segment fields, and the segments were used as masks for the classification using Sentinel-1 images. The co-registration of Sentinel-1 and Sentinel-2 was achieved using GEE. In view of the cloud cover, Sentinel-2 images from June to August were selected by median filtering to composite an image of Erguna with low cloud coverage. Generally, the boundary of the field does not change much over the following year, so even if there are no optical images in this year, the image of the previous year can still be used to segment the fields.

The farmland managers in the study area provided detailed information on some fields from 2016 to 2019 for the study, including the longitude and latitude of fields, the crop types, the sowing dates and the agricultural management methods. The ground truth of the year 2019 is shown in Figure 1b, and Table 2 shows the fields' number of ground truths in different years. There are different management methods for the fields. Some fields in the study area were ploughed before sowing, and some fields were ploughed after harvest. In addition, pesticides were applied to some fields to control the vigorous growth of crops.

**Table 2.** Number of ground truth fields provided by farmland managers in different years.

|  |  | 2016 | 2017 | 2018 | 2019 |
|--|--|------|------|------|------|
| Wheat | Number of Fields | 206 | 252 | 223 | 255 |
|  | Area (ha) | 4551 | 4937 | 4873 | 5369 |
| Canola | Number of Fields | 97 | 147 | 190 | 137 |
|  | Area (ha) | 2767 | 2781 | 4335 | 2148 |

In situ measurements were performed from August 1 to 9 (fruit development stage) in the study area in 2017. The crop type, height, soil moisture and wet and dry weight of biomass were all measured for each crop. Due to the low precipitation and high temperature during the crop growth stage in 2017, as shown in Figure 3, the growth conditions of the crops were poorer than those in the three other studied years. The yields in Figure 4 also show the poor growth of the crops in 2017. Figure 5 shows the canola and wheat in different growth conditions in 2017. It shows that there is a large difference in height, biomass and canopy density between well- and poorly grown wheat fields, and the same for canola fields.

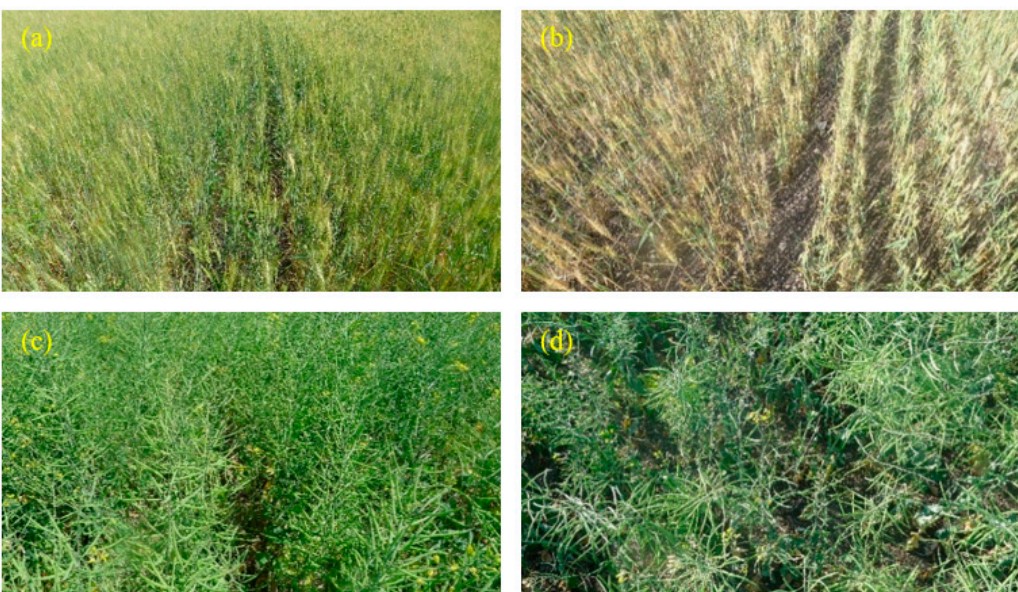

**Figure 5.** Canola and wheat with different growth conditions in 2017 in the ripening stage: Wheat (**a**) in good growth condition and (**b**) in bad growth condition. Canola (**c**) in good growth condition and (**d**) in bad growth condition.

## 3. Methodology

The methodology is shown in Figure 6. First, the Globeland30 data were used to extract the farmland. Then, the composited Sentinel-2 images were used to segment the farmland to obtain the field mask. Canola and wheat were then classified on single- or multi-temporal SAR images masked by the fields' segments by RF classifier in GEE. The importance scores of features were derived from the multi-temporal classification.

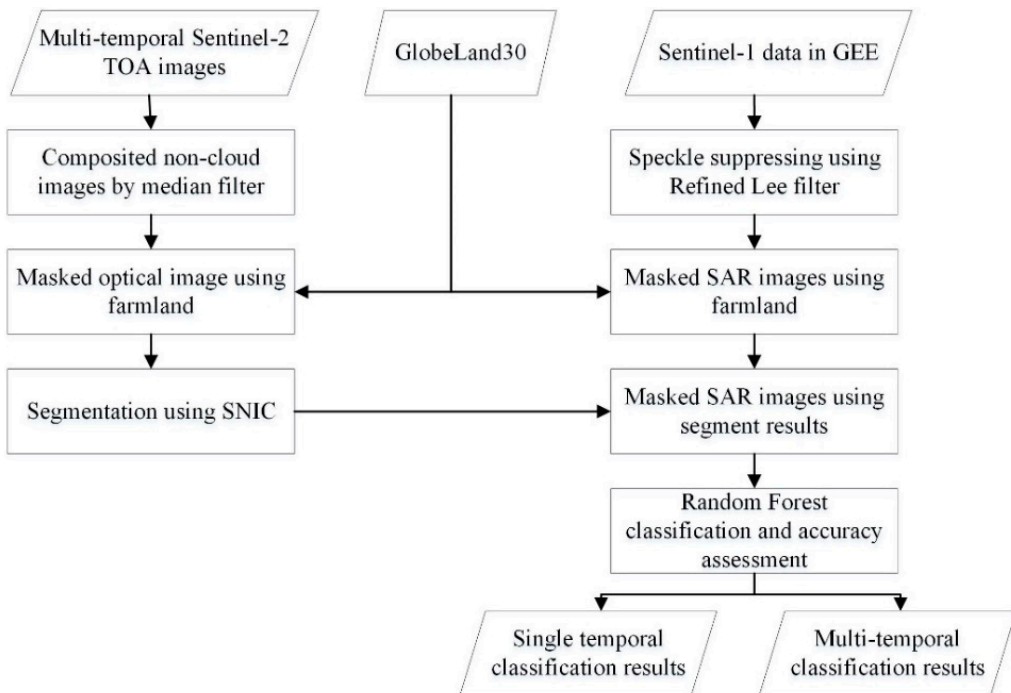

**Figure 6.** The flowchart of the methodology.

### 3.1. Segmentation of Fields by Sentinel-2 Images

The pixel-based SAR classification results are noisy because of the heterogeneity within the fields and the speckle noise induced by the coherent imaging of SAR images [3]. A feasible means is object-oriented classification [24], that is, pre-segmenting images and then classifying them. In this paper, the simple non-iteration cluster (SNIC) superpixel segmentation [37] was performed over a composited Sentinel-2 image to obtain the segments of the study area. The SNIC segmentation algorithm clusters pixels without the use of the K-means iterations, while explicitly enforcing connectivity from the start [37], which makes it computationally cheaper and uses less memory.

Median filtering was used in filtering the Sentinel-2 TOA images to obtain a composite image with a low cloud cover. Temporal images with a cloud coverage of less than 20% from July to August were selected for median filtering. From 2016 to 2019, the number of Sentinel-2 images used to construct cloudless images was 96, 96, 174 and 201, respectively. In view of the resolution of Sentinel-1 images, the visible bands, red edge band and infrared band of Sentinel-2 were selected, and then the median Sentinel-2 image was masked by farmland extracted from GlobeLand30. Finally, the segments of farmland were obtained using SNIC for each year, using the corresponding composited Sentinel-2 image.

### 3.2. Random Forest Classification and Accuracy Assessment

The RF classifier is an efficient classification method developed from ensemble learning and decision trees [38]. Meanwhile, the importance scores of input features can be derived from RF, which enable the assessment of contribution of each feature to the classification results. In this paper, Gini importance [39] was used to assess the contribution of features in the classification. It calculates each feature importance as the sum over the number of splits (across all tress) that include the feature, proportional to the number of samples it splits. The input features for the RF classifier are the backscattering coefficients of VH ($\sigma_{VH}$), VV ($\sigma_{VV}$) and the cross-pol ratio ($\sigma_{VH}/\sigma_{VV}$). In this paper, the number of trees of RF was set at 200, the number of features per split was the square root of the number of features, the fraction of samples input to bag per tree was 0.5 and the minimum samples of leaf nodes was 1. A certain number of samples were randomly selected from the ground truth of each year. A total of 30% of the ground truths were used as training samples, and the remaining 70% were used as validation samples.

The producer accuracy (PA), user accuracy (UA), overall accuracy (OA), kappa coefficient and F1 score [40] were used to assess the classification results. The kappa coefficient can make the accuracy assessment comprehensive by reducing the effect of uneven sample distribution. The F1 score is the harmonic average of PA and UA (Equation (1)), reflecting the overall performance of the classification results [41]. In this experiment, $\beta$ was set at 1, indicating that PA and UA are of equal importance.

$$\text{F1} = \frac{(1 + \beta^2) \times \text{PA} \times \text{UA}}{\beta^2 \times (\text{PA} + \text{UA})} \tag{1}$$

## 4. Results and Analysis

In this section, the temporal backscattering profiles of wheat and canola in four years are compared, and the single- and multi-temporal classification results are analyzed to show the discrimination of wheat and canola at different phenological stages and in different years. The segmentation results using composited Sentinel-2 images of 2019 are shown in Figure 7. Patches in different colors represent the objects after segmentation. The segmentation considers the Euclidean distance of the pixels, and the segmented object tends to be a square shape. The used Sentinel-1 images were masked based on the segmentation results.

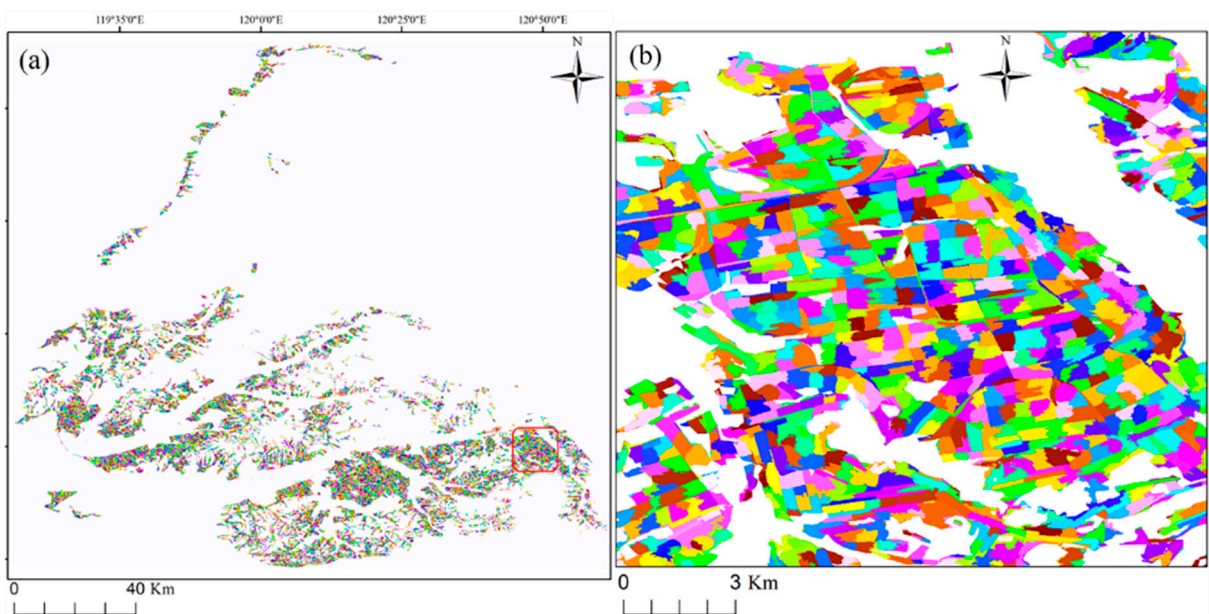

**Figure 7.** (**a**) Segmentation results using composited Sentinel-2 images of 2019 and the red box is (**b**) the zoomed-in area. Patches in different colors represent the objects after segmentation.

### 4.1. Backscattering Analysis of Canola and Wheat

In the following sections, the temporal characteristics of the backscattering of canola and wheat are analyzed. The mean value and standard deviation of the backscattering coefficients derived from all the canola and wheat fields in ground truth from 2016 to 2019 are shown in Figure 8. As the figure shows, the backscattering profiles of one crop are different in the four years. The fluctuations over the years hinder the finding of a stable indicator for crop monitoring using SAR images, because the radar backscattering of crops is affected by many factors, including crop status and soil moisture.

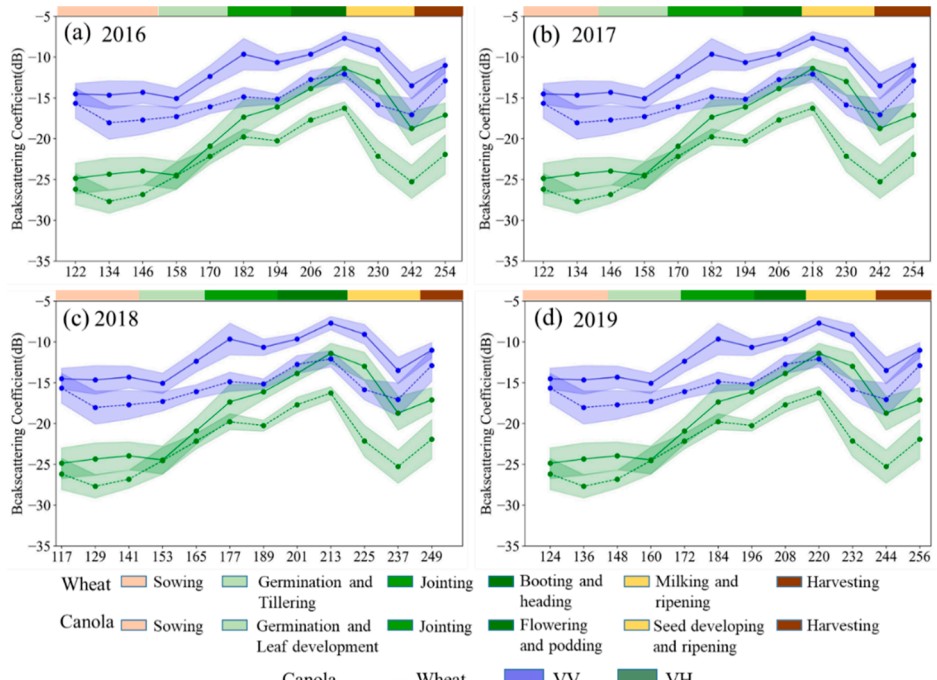

**Figure 8.** Temporal backscattering profiles of canola and wheat in the year of (**a**) 2016, (**b**) 2017, (**c**) 2018 and (**d**) 2019. Dots represent mean values, and fill colors represent the standard deviations of canola and wheat parcels.

Generally, $\sigma_{VV}$ is higher than $\sigma_{VH}$ at all the growth stages of canola and wheat, except at the sowing stage, when the X-Bragg scattering is the dominant scattering mechanism [42]. The largest difference in backscattering was observed for the four years at the sowing stage, when the ground roughness and soil moisture are the main factors for backscattering in different years. After the tillering and leaf developing stage, the biomass and canopy structure play an increasingly important role in the backscattering. To understand the backscattering changes in canola and wheat more clearly, the growth profiles of canola and wheat over the studied period are analyzed as follows.

### 4.1.1. Backscattering Profiles of Canola

The multi-year temporal profiles of canola are shown in Figure 8. In different years, although $\sigma_{VV}$ was always stronger than $\sigma_{VH}$, $\sigma_{VH}$, which has a strong correlation with biomass, had greater changes than $\sigma_{VV}$ during the growth cycle. In the sowing and germination stages, canola has low biomass, and the SAR signal was mainly affected by the soil moisture and surface roughness. From the end of May to the beginning of July, canola develops from the germination stage to the jointing stage, and the $\sigma_{VH}$ of canola showed a gradually increasing trend as a result of its increasing biomass. The subsequent enhancement of $\sigma_{VH}$ can be explained by the appearance of flowers and pods, which form a dense and wet layer in canopy. Starting in late July (the podding stage), the variation in $\sigma_{VH}$ started to decrease due to its smaller structural change and the weaker effect of the underlying surface than that in the earlier stages. This is similar to the observation of Deschamps et al. [43]. Canola develops pods and becomes mature in August (the podding and fruit development stage), and in the analysis, the biomass peaked, which made the $\sigma_{VH}$ fluctuate less and peak at the podding and fruit development stage, peaking around $-10$ dB in all four years. However, $\sigma_{VH}$ saturated from the development of fruit to the ripening stage due to a dense canopy formed by canola pods and the limited penetration of the C-band wave. A noticeable drop was observed at the beginning of September (the harvesting stage), when canola was in seasonal senescence and harvesting. However, the backscattering coefficients were still higher than those in the sowing period due to the residues [42].

The changes in $\sigma_{VV}$ were more complicated than those in $\sigma_{VH}$ because $\sigma_{VV}$ is more sensitive to the differential attenuation than $\sigma_{VH}$. From the germination stage in late May to the jointing stage in early July, the $\sigma_{VV}$ of canola increased by 5–6 dB in all years. While the $\sigma_{VV}$ increased slightly from the jointing stage in July to the ripening stage in August, the increase was about 2 dB except for the canola in 2019. Although there was a sharp increase in the $\sigma_{VV}$ of canola from the leaf developing to the jointing stage in 2018, making a peak in $\sigma_{VV}$ in late June of 2018, $\sigma_{VV}$ reached the subsequent peak in the fruit development stage in August at about $-7$ dB, which is close to the peaks in other years. The backscattering profiles in recent years show that the growth profiles of canola are stable during the flowering and fruit development stages.

### 4.1.2. Backscattering Profiles of Wheat

Wheat exhibits a relatively lower backscattering than canola in $\sigma_{VV}$ and $\sigma_{VH}$, as shown in Figure 8. The main reason is that the biomass of wheat is usually lower than that of canola, and wheat also has a significantly vertical differential attenuation after the jointing stage [28,30]. In addition, the multi-year temporal profiles of wheat had larger changes than those of canola, because the relatively sparse canopy of wheat allows for deep penetration of the C-band wave. The scattering from the soil and the interaction of microwave with wheat changes in different growth conditions cannot be ignored. Different from canola, the $\sigma_{VH}$ of wheat peaked earlier. Except for 2019, the $\sigma_{VH}$ of wheat in other years had a small change at the jointing and ripening stages for the relatively similar structure of wheat, and the mean value of $\sigma_{VH}$ fluctuated around $-20$ dB. The differences in biophysical parameters between wheat and canola were greater during this period compared with those in earlier periods, creating a large difference in the backscattering between wheat and canola at the

stage. Similar to canola, wheat began to senesce after maturity in late August, leading to a decrease in the water content of the wheat canopy. The only exception was in 2017, when the backscattering of VH still remained stable at the senescence and harvesting stage, which may be due to the difference in harvesting time and the post-harvest management. Compared with $\sigma_{VH}$, the $\sigma_{VV}$ of wheat was mainly the direct scattering from soil and the attenuation caused by the predominately vertically oriented wheat stems, and it varied considerably from year to year.

### 4.2. Classification Results of Canola and Wheat on Single-Temporal Images

The classification results of the crops in single-temporal images acquired in 2019 are shown in Figure 9. The zoomed-in area shows the Yigen Farmland, where there was ground truth provided by farm managers, and the results show that canola and wheat were well classified in different phenological stages. Figure 10 shows the classification accuracies for different images acquired in 2019. The lowest OA was 84%, which was obtained on 16 May, corresponding to the germination stage of wheat and canola, and the highest OA of, 96%, was reported from 27 July to 20 August, corresponding to the podding and ripening stage of canola. The highest accuracy was also consistent with the earlier phenological analysis: that the backscattering of canola and wheat has the largest difference and stability during the flowering to ripening stages.

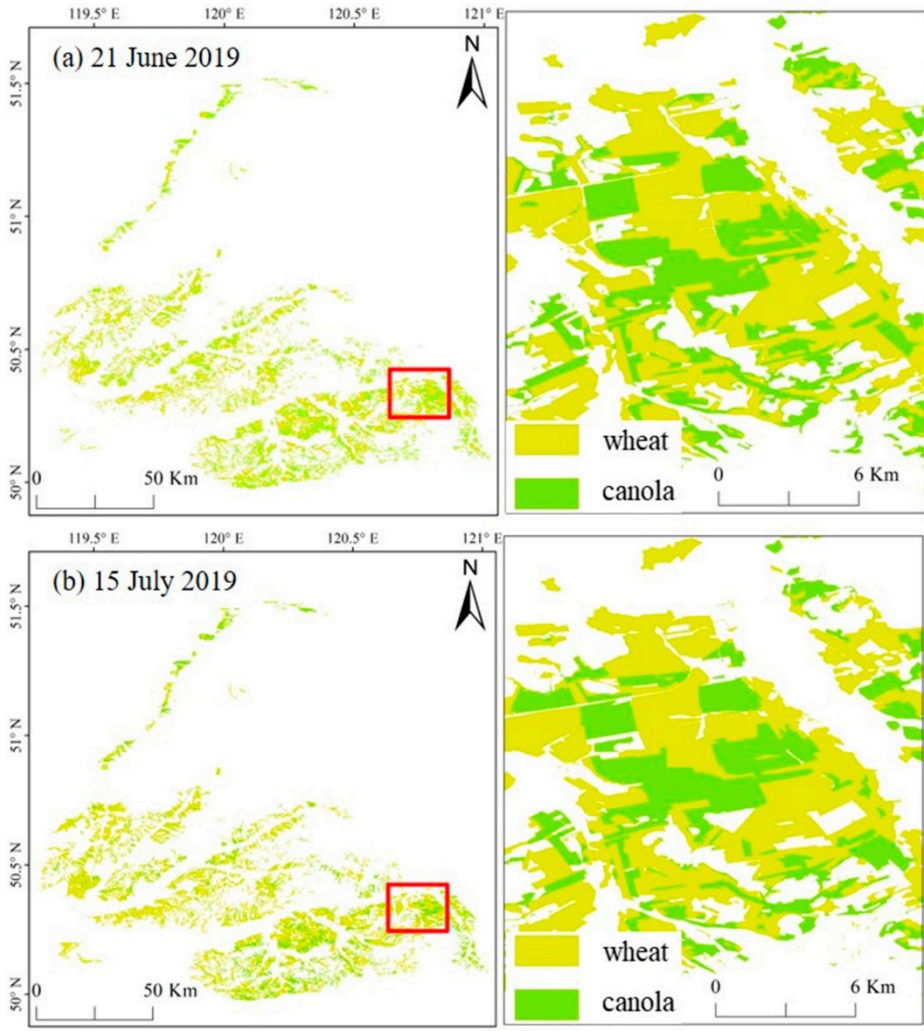

**Figure 9.** *Cont.*

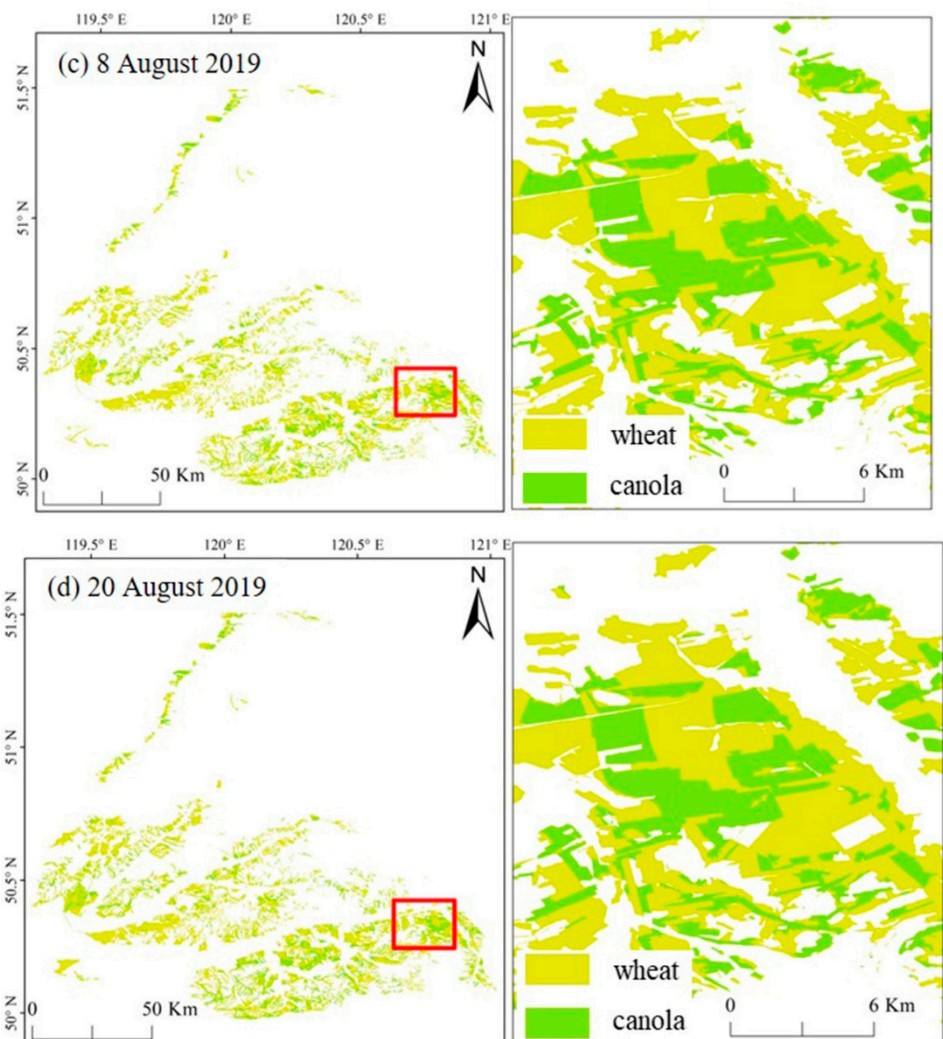

**Figure 9.** Classification results based on single-temporal SAR images acquired on (**a**) 21 June 2019: Jointing stage, (**b**) 15 July 2019: Canola flowering stage, (**c**) 8 August 2019: Seed developing stage and (**d**) 20 August 2019: Ripening stage. The figures in right are the zoomed in area shown in red box.

The mapping accuracies of canola were high in the middle periods and significantly decreased at the harvesting stage. The F1 score of canola before leaf development was lower than that after leaf development, and the F1 score of canola reached the maximum value of 0.98 on 27 July. From 15 July to 20 August, canola achieved satisfying mapping accuracy, and the F1 scores remained above 0.95. Their classification accuracies decreased below 0.9 at the beginning of September due to complex harvest patterns [42].

During the whole growth cycle, the PAs of wheat were above 90%, and the maximum value was obtained on 15 July. Similar to canola, UA gradually increased with the growth of wheat, and the gap between the UA and the PA of wheat narrowed because the structural differences in wheat canopy in the middle and late stages were insignificant. However, there was a large difference in soil moisture and roughness and sowing time for the fields in the early stages. The F1 scores of wheat remained above 0.85 in all periods and exceeded 0.95 from 3 July to 20 August, corresponding to the heading, flowering and milking stages.

The classification accuracies are strongly related to the phenology of canola and wheat, which is consistent with the backscattering variation mentioned in previous section. In all the four years, the greatest differences in backscattering between canola and wheat were found in July and August, corresponding to the flowering, milking and ripening stages. The classification accuracy corresponding to the above period was also higher than that of other periods. The results show that the best stages for canola and wheat classification are

the flowering and milking stages. Although the separability of canola and wheat varies in different phenological stages, the Sentinel-1 data generally achieved good separability for canola and wheat in all stages.

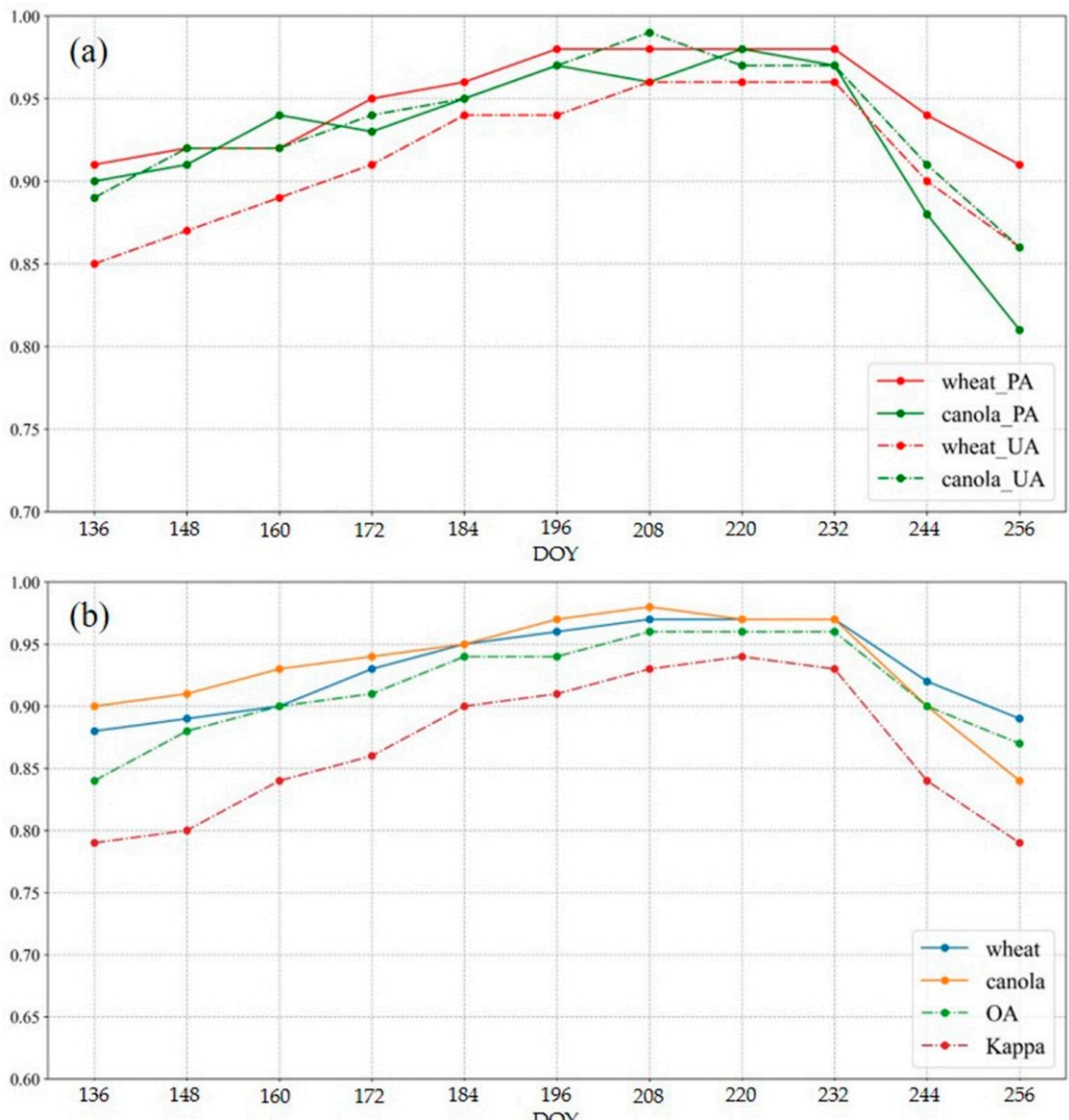

**Figure 10.** Classification accuracies based on single-temporal Sentinel-1 SAR data in 2019. (**a**) Producer accuracy and user accuracy; (**b**) F1 score, overall accuracy and Kappa.

### 4.3. Classification Results of Canola and Wheat Using Multi-Temporal Images

The temporal profiles in Figure 8 show the fluctuations in backscattering caused by agriculture management and agrometeorological conditions in different years. To verify the mapping capability of canola and wheat using temporal Sentinel-1 dual-pol GRD data, images in Table 3 that were acquired in each year (2016–2019) were used for crop classification. Figure 11 shows the mapping results of canola and wheat in different years. The classification accuracies are shown in Table 3, which shows that the F1 scores of canola and wheat in four years were all higher than 0.9. The results show the strong discrimination capability of temporal SAR images on wheat and canola, which have significantly different canopy structures. Although Figures 4 and 5 show that there was a large difference in growth conditions for different years or different fields in the same year, the different growth conditions did not have a large effect on the classification of canola and wheat.

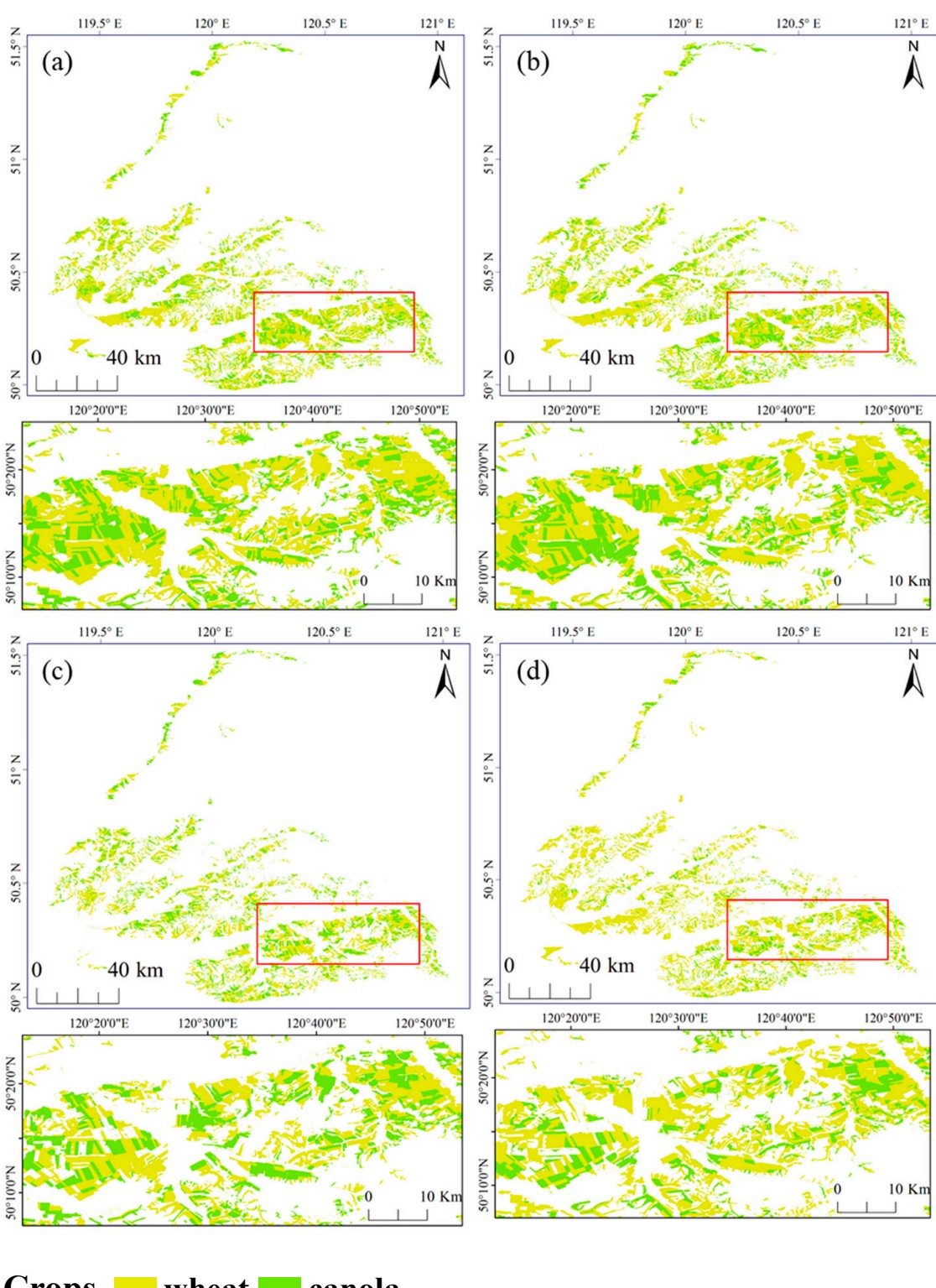

**Figure 11.** Multi-temporal classification results of Erguna in (**a**) 2016, (**b**) 2017, (**c**) 2018 and (**d**) 2019, and the same zoomed-in areas of the region in red box.

**Table 3.** Classification accuracies of wheat and canola using multi-temporal images in different years.

| Year | | Wheat | Canola | OA | Kappa |
|---|---|---|---|---|---|
| 2016 | PA | 0.96 | 0.91 | 0.94 | 0.87 |
| | UA | 0.94 | 0.92 | / | / |
| | F1 | 0.95 | 0.92 | / | / |
| 2017 | PA | 0.94 | 0.97 | 0.95 | 0.90 |
| | UA | 0.98 | 0.90 | / | / |
| | F1 | 0.96 | 0.93 | / | / |
| 2018 | PA | 0.95 | 0.97 | 0.95 | 0.93 |
| | UA | 0.95 | 0.97 | / | / |
| | F1 | 0.95 | 0.97 | / | / |
| 2019 | PA | 0.96 | 0.95 | 0.96 | 0.94 |
| | UA | 0.98 | 0.93 | / | / |
| | F1 | 0.97 | 0.94 | / | / |

### 4.4. Feature Importance Analysis

The relative importance scores of features were used to show the contributions of $\sigma_{VH}$, $\sigma_{VV}$ and cross-pol ratio (R), as shown in Figure 12. The importance of each feature changed with the phenological stages. Figure 12 demonstrates that features in July and August had higher scores than that in other dates for all years. This finding is also consistent with the backscattering characteristic analysis and single-temporal classification results: There is a large difference in the backscattering of canola and wheat from the flowering stage to the ripening stage, and the classification results during these stages achieved high accuracy. The importance scores of $\sigma_{VH}$ and $\sigma_{VV}$ for canola and wheat mapping did not have a large differences for one stage. Thus, the dual-pol channels all had a strong contribution to the separability of canola and wheat. The cross-pol ratio can weaken the double bounce between soil and crop stalks, and has been used as a stable indicator in the phenological stage monitoring of crops [33,44]. However, its importance scores in the classification of wheat and canola was not always high. Given that the phenological-stage monitoring pays more attention to the crop canopy and tends to obtain a stable indicator, the canopy–ground interaction or the effect of different management methods may favor the classification of wheat and canola to some extent. Figure 12 shows that the dual-pol backscattering coefficients were enough for the mapping of wheat and canola, especially from the jointing to the ripening stage. However, this does not mean that features with a low score cannot be used for the classification of canola and wheat, because there is some information redundancy between the features. In other words, if two features are highly correlated and one feature has a high score, the other feature may have a low score.

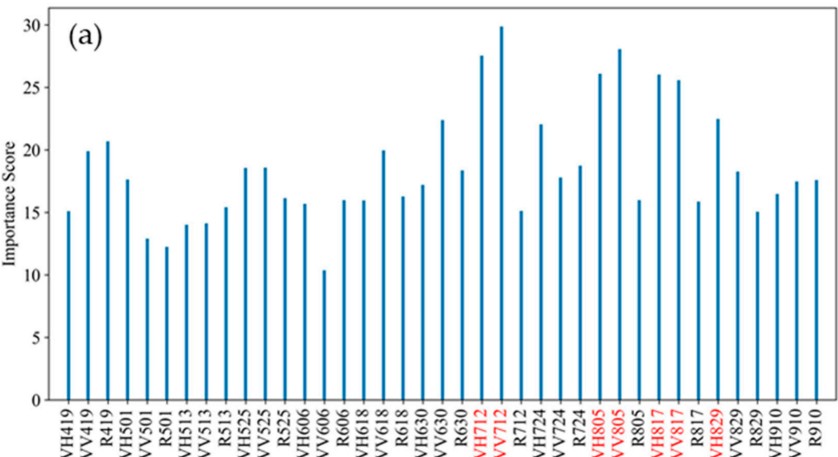

**Figure 12.** *Cont.*

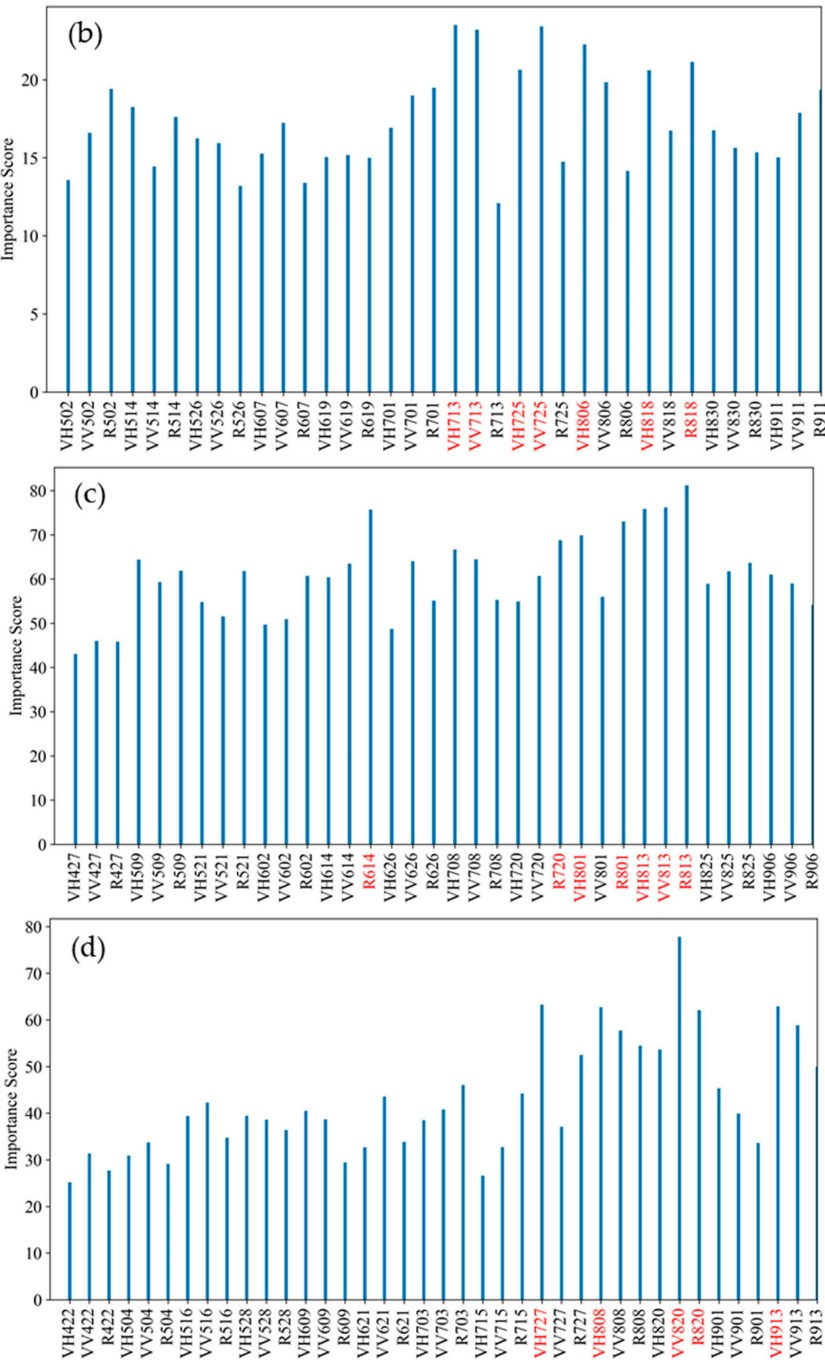

**Figure 12.** Feature importance scores derived from the random forest classification for the year (**a**) 2016, (**b**) 2017, (**c**) 2018 and (**d**) 2019, and the features in red have higher scores than the other features.

## 5. Conclusions

This paper investigates the temporal characteristics of canola and wheat in different years and evaluates the capability of Sentinel-1 data for the mapping of canola and wheat when the multi-year radar backscattering profiles have large fluctuation. The fluctuations were induced by agrometeorological conditions and field managements in different years, and made it difficult to find the standard growth profiles for canola and wheat. Larger fluctuations were observed for canola and wheat over the early stages when the canopy was sparse, and over the harvest stage when the harvest patterns were complex. Low variances in backscattering were observed for canola and wheat from the flowering to the

ripening stage, when high biomass limited the penetration, and the canopy of fields with different sowing dates became similar.

Although there were large fluctuations for the multi-year backscattering of canola and wheat, the different canopy structures of wheat and canola made them have significantly different backscattering coefficients. The classification results show that there was good separability for canola and wheat in all stages, and canola and wheat have a wider time window for good-quality mapping with Sentinel-1 images than that with optical or multi-spectral images. This finding is also consistent with the single-temporal classification results and the importance scores of multi-temporal features. The single-temporal classification accuracies of canola and wheat in the jointing stage were close to 90%, which is useful for their mapping in the early phenological stage. The similar accuracies of single- and multi-temporal classification in the middle phenological stages indicate that the single-temporal Sentinel-1 image can sufficiently map the canola and wheat well. These backscattering characteristics analysis and classification results show the capability of Sentinel-1 SAR images in the mapping of canola and wheat on a large scale. In the future, canola and wheat in different phenological divisions will be mapped in GEE using SAR images.

**Author Contributions:** Conceptualization, L.Z., W.S. and S.W.; methodology, L.Z., S.W., Y.X. and J.Y.; validation, S.W., L.Z. and Y.X.; formal analysis, L.S. and J.D.; resources, L.Z.; writing—original draft preparation, S.W., L.Z. and writing—review and editing, L.Z. and J.D.; visualization, S.W.; Supervision, L.Z., J.Y. and W.S.; project administration, L.Z.; funding acquisition, L.Z., J.Y. and L.S. All authors have read and agreed to the published version of the manuscript.

**Funding:** This research was funded by the National Natural Science Foundation of China [grant No. 61971318, No. 42001134, No. U2033216], the Natural Science Foundation of Hubei Province [grant No. 2022CFB193] and the Shenzhen Fundamental Research Program [grant No. JCYJ20200109150833977].

**Data Availability Statement:** The SAR datasets created during the study are available at the Copernicus Open Access Hub (https://scihub.copernicus.eu/, accessed on 1 January 2023) or in Google Earth Engine.

**Acknowledgments:** We would like to thank the farmers in the Yigen farmland for providing the ground measurements and yields of crops in different years. In addition, we would like to thank the scientists in the Chinese Academy of Forestry for providing part of the in situ measurements for the study.

**Conflicts of Interest:** The authors declare no conflict of interest.

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
