# Peer review of "Evaluating the Capability of Sentinel-1 Data in the Classification of Canola and Wheat at Different Growth Stages and in Different Years"

_remotesensing, doi:10.3390/rs15112731_

Round 1
Reviewer 1 Report
The manuscript analyzes the characteristics and their differences between canola and wheat on multi-temporal Sentinel-1 SAR data. It uses single temporal and multi temporal Sentinel-1 data to investigate the optimal time window for their classification. The results illustrate the wide time window for the classification of wheat and canola because of their large different canopy structure. It is interesting to illustrate the time window for the manuscirpt using multi-year images when wheat and canola often have different growth conditions. The manuscript is well structured. However, there are still some details that need attention, as follows:
1) Some spelling errors in the manuscript need to be corrected, like the unit of area in Sec.2.1 should be “km2”, not “km2”.
2) The growth calendar of wheat and canola should be separated to distinguish the difference between them.
3) It is suggested to unify the expression of dates in the manuscript, such as using Day of Year or using month and day for all dates.
4) Have you considered co-registration of images as multi source images are utilized
5) Why the segmentation result is square? Please clarify it.
6) Typically, 30% of the sample is used for training and 70% for verification, but the opposite is true in the manuscript. Please check for any errors.
7) In the flowchart, the “refined Lee” should be “Refined Lee”.
8) The y-coordinate information in Figure 3 is incorrectly marked.
9) Line 48,49, Grammatical errors.
In the paper , the expression may not be authentic English. It is suggested the english language should be revised.
Reviewer 2 Report
This paper investigates the temporal characteristics of canola and wheat in different years and evaluates the capability of Sentinel-1 data for the mapping of canola and wheat when the multi-year radar backscattering profiles have large fluctuations. However, there are still some deficiencies that need improvement.
1. The analysis of the backscatter coefficient and classification results is not well integrated with the phenology of crops. It is recommended to add more analysis in Session 4.
2. Why choose the research area in Inner Mongolia? This is not the main wheat-producing area.
3. Check the grammar of the manuscript. Some hyphens were incorrectly used.
4. The images in the manuscript are not clear enough. And some words in the image have too small fonts.
Check the grammar of the manuscript. Some hyphens were incorrectly used.
Reviewer 3 Report
The authors used time-series Sentinel-1 SAR data to classify canola and wheat crops in Erguna, Inner Mongolia, China, using the GlobeLand30 dataset to mask farmlands and farmland manager information as ground truth. Before classification, they used Sentinel-2 images with the simple non-iteration cluster (SNIC) algorithm in Google Earth Engine for super-pixel segmentation. They used the random forest classifier and achieved the highest classification accuracies of 96% during the canola flowering and podding stages due to stability in backscattering profiles during those stages. The knowledge gap is clearly stated, methods are appropriate and clearly described, and the results are explained in detail in the discussion. Questions and comments follow.
1) One more read-through for grammatical corrections would be useful.
2) Would the workflow work if more than canola and wheat were included in the classification?
3) How well would the workflow transfer to other areas and/or crop types?
Minor editing of English language required
Reviewer 4 Report
The paper investigates the backscattering profile of Canola and Wheat crops derived from Sentinel-1 time-series images to 1) highlight their main evolution trends and fluctuations, 2) explore the SAR capability for the multi-temporal classification of these crops in a large area of Northeastern China.
The Manuscript is well-structured and clearly presented. The methods used are suitable for the study’s aims. The Authors could provide additional information in the last section of the paper (4.4 Feature importance analysis), also improving the readability of the x-axis caption of the graph in Fig 12. Furthermore, I have very few minor suggestions, which can help the Authors to improve the Manuscript quality:
1. Row 92: correct km2 to km2
2. Row 96-97: “planted in late April” or “sown from April” ?;
3. Figure 3: invert the y-axis captions;
4. Table 1 – page 4: “24-Jul” instead of “24-Jun”;
Regarding the use of kappa for accuracy assessment, although this index has been intensively used in the past, currently it would be appropriate to better specify the reasons that led to its use in the article.
Reviewer 5 Report
Crop identification by using RF classifier is really an interesting topic but I think that some statistical results are really missed in this research. In particular, plots show only S-1 average backscatter values without any information concerning standard deviation, for instance, or other statistical parameters that could strongly improve the trends analysis by providing variability inside classes.
I suggest reconsider the connections between sentences to make them, as general suggestion, a bit longer. Text, as it was presented, appears quite fragmented.
Round 2
Reviewer 5 Report
The corrections and integrations that were asked after the first revision have been correctly finalized as well English structure that has been improved. The plots showing the results are now definitely more interesting after having added statistical information. No additional changes are required.
Author Response
Thank you for the time dedicated to review the revised manuscript and for the constructive comments and valuable suggestions.